# Best Practices on Radiology Department Workflow: Tips from the Impact of the COVID-19 Lockdown on an Italian University Hospital

**DOI:** 10.3390/healthcare10091771

**Published:** 2022-09-14

**Authors:** Fabio Pellegrino, Aldo Carnevale, Riccardo Bisi, Davide Cavedagna, Roberto Reverberi, Licia Uccelli, Stefano Leprotti, Melchiore Giganti

**Affiliations:** 1Department of Translational Medicine, University of Ferrara, 44124 Ferrara, Italy; 2Department of Radiology, Sant’Anna University Hospital Ferrara, 44124 Ferrara, Italy; 3Blood Transfusion Service, Azienda Ospedaliera Universitaria di Ferrara, 44124 Ferrara, Italy

**Keywords:** radiology, COVID-19, emergency radiology, radiology department, trauma, lean healthcare

## Abstract

Purpose: The workload of the radiology department (RD) of a university hospital in northern Italy dramatically changed during the COVID-19 outbreak. The restrictive measures of the COVID-19 pandemic lockdown influenced the use of radiological services and particularly in the emergency department (ED). Methods: Data on diagnostic services from March 2020 to May 2020 were retrospectively collected and analysed in aggregate form and compared with those of the same timeframe in the previous year. Data were sorted by patient type in the following categories: inpatients, outpatients, and ED patients; the latter divided in “traumatic” and “not traumatic” cases. Results: Compared to 2019, 6449 fewer patients (−32.6%) were assisted in the RD. This decrease was more pronounced for the emergency radiology unit (ERU) (−41%) compared to the general radiology unit (−25.7%). The proportion of investigations performed for trauma appeared to decrease significantly from 14.8% to 12.5% during the COVID-19 emergency (*p* < 0.001). Similarly, the proportion of assisted traumatic patients decreased from 16.6% to 12.5% (*p* < 0.001). The number of emergency patients assisted by the RD was significantly reduced from 45% during routine activity to 39.4% in the COVID-19 outbreak (*p* < 0.001). Conclusion: The COVID-19 outbreak had a tremendous impact on all radiology activities. We documented a drastic reduction in total imaging volume compared to 2019 because of both the pandemic and the lockdown. In this context, investigations performed for trauma showed a substantial decrease.

## 1. Introduction

Since December 2019, severe acute respiratory syndrome coronavirus 2 (SARS-CoV-2) has rapidly spread worldwide, and WHO characterized the coronavirus disease 2019 (COVID-19) outbreak as a pandemic on 11 March 2020 [1]. Italy was one of the countries most affected by the pandemic during the first months of that year. On 8 March 2020, the Italian government implemented extraordinary measures to limit viral transmission that were intended to minimise the likelihood of contact between people who were infected with people not infected [2]. Emilia-Romagna, together with other regions in northern Italy, was severely hit with a count of 28,869 confirmed cases and 4269 deaths by 11 July 2020 [3]; however, only 1079 people had been infected in the province of Ferrara, which demonstrated a limited impact in this specific subarea [4], mainly due to heterogeneous local characteristics that may have influenced the contagion risk [5].

While the central Italian government had adopted strict measures to limit the spread of the infection in the population—measures culminating with the establishment of the national lockdown starting on 11 March 2020 [6]—the Italian public healthcare system was overloaded by the abrupt surge of COVID-19 cases [7].

In these circumstances, there was a significant redefinition of the health services offered to the population by hospitals and territorial institutions due to the reduction of most routine activities and the subsequent diversion of a considerable amount of resources towards the management of the pandemic. This situation, concurrently with movement restriction issued by the central government and the widespread fear of contagion, led to a decrease in the access to emergency departments and hospitalizations related to several serious pathologies, such as stroke or metabolic diseases [8,9,10,11,12,13]. Radiology departments (RDs) had to adapt to this situation to maintain an adequate level of radiological and interventional support, while ensuring the ability to face a crisis of unexpected proportions [14]. Changes to working protocols and procedures were implemented to limit the spread of the virus between patients in hospital wards and to guarantee the safety of healthcare workers [15,16,17,18]. Although nowadays, the number of emergency department (ED) admissions for trauma has increased with an increment in the use of diagnostic technology due to its greater availability [19], it is unclear the impact of lockdown restrictions and the perception of hospitals during a pandemic on diagnostic tests for trauma. To improve the management of radiological services in the COVID-19 pandemic, the health management model called lean healthcare is an ideal candidate for optimizing the work processes. Its tools have been applied in this pandemic scenario, helping the performance, facing challenges, such as staff shortages and rising costs, and providing high-quality services, while also considering the current financial restrictions [20]. As evidenced by the literature, implementing lean healthcare offers remarkable opportunities to improve overcrowding, costs, patient flow, difficulty in screening, and elimination of inefficient processes (i.e., waiting, movement, and overprocessing) [20].

Sharing the experiences from strained healthcare systems is essential in understanding how to achieve the highest possible health and safety levels and how to preserve and optimize health resources. This article reports the response of the RD of an academic hospital during the first COVID-19 outbreak in the Emilia-Romagna region, intending to describe how the workload of the RD changed in this dramatic scenario and to assess the variation in the diagnostic activity linked to trauma cases.

The purpose of this study was, in conclusion, to suggest, according to the data collected in our experience during the acute phase of the pandemic, how to improve the workflow of RDs based on lean healthcare goals, such as the identification and elimination of inefficient processes in future similar pandemic scenarios regulated by social distancing efforts, such as shelter-in-place, and other containment measures.

## 2. Methods

### 2.1. University Hospital “Arcispedale Sant’Anna”

*Arcispedale Sant’Anna* in Ferrara (Emilia-Romagna, Italy) is a 711-beds university hospital with a catchment area of about 350,000 people. The RD provides imaging services for hospitalized, day-hospital inpatients and for outpatients; it also grants assistance to the ED.

The working area available to the RD comprises three sectors on the ground floor of the structure: an emergency radiology unit (ERU) located near the ED and a general radiology unit (GRU), while a separate sector is reserved for interventional radiology. The senology unit is in a separate subunit.

The ERU works twenty-four hours a day, seven days a week to support the activities of the ED, ensuring X-ray, ultrasound, and CT investigations for medical and surgical emergencies. ERU staff handles not only ED patients but also two specific categories of hospitalized patients, namely those admitted to the emergency medicine unit—which also houses patients under intensive, short-term observation—and to the emergency surgery unit.

The GRU ensures imaging for hospitalized and day-hospital inpatients, as well for outpatients.

### 2.2. Data Source and Population

Data on the diagnostic services performed during the COVID-19 emergency from 9 March 2020 to 26 May 2020 by the RD of our university hospital were collected through the RIS-PACS (integrated radiology information system-picture archiving communication system).

Data was analysed in aggregate form and compared with those related to the same timeframe of the previous year from 11 March 2019 to 26 May 2019.

Since there were no acquisitions of new machinery or decommissioning of scanners and since the comparison between the number of examinations performed in 2018 and 2019 showed a relatively small difference of +4.53% in 2019, the span between 11 March 2019 and 26 May 2019 was considered an appropriate reference for the routine workflow of the department.

For the assessment of prevalence, the following investigations in the two timeframes were analysed:-X-rays examinations, including bedside ones, performed respectively by the ERU and GRU;-Ultrasound investigations, including bedside ones, performed respectively by the ERU and GRU;-CT scans conducted respectively by the ERU and GRU;-MRI scans performed by the GRU.

Data were sorted by patient type in the following categories: inpatients, outpatients, and ED patients; furthermore, patients were defined according to the diagnostic path based on history and symptoms. This classification was implemented by analysing the “clinical picture” and “clinical question” fields of the data extracted from the RIS-PACS.

To identify examinations performed following a trauma, the “clinical picture” and “clinical question” fields were analysed. An investigation was deemed to have been performed on a trauma case if at least one of the following conditions were identified in the data:-trauma;-accidental or syncopal fall;-road accident;-wound (incised wound, laceration, abrasion, penetration wound, avulsion, traumatic amputation);-crush injury;-hematoma;-assault or scuffle;-distractive injuries and strain;-violent suicide attempt (hanging);-excessive physical exertion during work or sporting activity.

A patient was deemed as “not traumatic” when one of the two PACS fields clearly stated the absence of reported trauma or no trauma signs.

Diagnostic and interventional procedures performed by the senology unit and interventional procedures performed by the GRU were excluded.

### 2.3. Statistical Analysis

The studied period comprises 11 weeks. The data of the two-time intervals were compared, both by considering the entire period and by dividing it into weeks. Data by period were compared with the chi-squared test on 2 × 2 or 2 × n tables and for the latter when the overall test showed a statistically significant difference (*p* < 0.05); a post hoc analysis was performed to establish which term of the comparisons was responsible for the difference found.

Weekly data were compared using the paired samples t-test or the paired samples Wilcoxon’s test, as appropriate.

A generalized linear model (Poisson regression) was employed, with year, patient type, and trauma as independent variables, and either number of examinations or number of different patients as the dependent variable. The statistical significance limit was set to 0.05, but in the case of multiple comparisons—for example, for the post hoc analysis of the 2 × n tables—values obtained were adjusted according to the Hommel procedure. The employed software was: Statistical Package for the Social Sciences (SPSS), version 19 (IBM Corp., Armonk, NY, USA) and Jasp version 0.16.3 [21].

## 3. Results

### 3.1. Variation in the Number of Examinations and Assisted Patients

Compared to the same period of 2019, between 11 March 2020 and 26 May 2020, 15,787 fewer examinations (−31.2%) were performed in the RD of our institution. The ERU registered a more marked reduction (−40.4%) in comparison to the GRU one (−25.5%). The proportion of examinations performed in the ERU decreased significantly from 38.3% of the total department diagnostic activity in normal operating conditions to 33.2% in the COVID-19 timeframe (*p* < 0.001).

Compared to the same timeframe of 2019, between 11 March 2020 and 26 May 2020, 6449 fewer patients (−32.6%) were assisted in the RD. This decrease was more pronounced for the ERU (−41%) compared to what occurred for the GRU (−25.7%). The proportion of patients assisted in the ERU decreased significantly: from 45% during routine activity to 39.4% during the COVID-19 emergency (*p* < 0.001). Data are presented more extensively in Table 1.

A statistically significant decrease was found in the number of diagnostic investigations performed per week in the COVID-19 emergency timeframe (*p* < 0.001) compared to the usual operating period (Table 2 and Figure 1).

Similarly, a significant decrease in the number of patients assisted by the RD was registered during the 11 weeks taken as reference. (*p* < 0.001) (Table 3 and Figure 2).

### 3.2. Variation in Diagnostic Activity by Patient Type and Method

Regarding the examinations by patient type, during the reference timeframe, 4926 fewer investigations (−32.5%) were performed on outpatients compared to the corresponding period of the previous year, while 5400 fewer (−27.3%) were performed on internal patients and 5461 fewer (−34.9%) performed on ED patients. The distribution of tests performed on the three categories of patients changed significantly in 2020 (*p* < 0.001). A post hoc analysis shows that during the COVID-19 emergency period, the proportion of examinations performed on ED patients decreased, while the proportion of investigations on internal patients increased.

Focusing on patients rather than on methods, the comparisons between the COVID-19 emergency period and the corresponding timeframe of the previous year shows that the for the RD assisted, there were:-31.2% fewer outpatients (−2264);-30.7% fewer inpatients (−1512);-38.7% fewer ED patients (−3106).

The distribution of the three patient types changed significantly between the two periods (*p* < 0.001). More specifically, after a post hoc analysis, it emerged that ED patients decreased in proportion to the other two types, which instead recorded an increase. Table 4 presents data in their entirety; data analysed by method are reported in the Appendix A.

### 3.3. Change in Diagnostic Activity Related to Trauma

Examinations for which it was not possible to establish with certainty whether they had been performed to investigate a trauma or a non-traumatic condition were excluded from the subsequent analysis: the excluded elements amounted to 92 for the 2020-time interval, and 121 for the 2019 period. Compared to the same period of 2019, it emerged that diagnostic services performed to investigate trauma cases between 11 March 2020 and 26 May 2020 decreased by 41.8% (−3122), while other diagnostic procedures decreased by 29.4% (−12,636) (Table 5). The proportion of investigations performed for trauma during the COVID-19 emergency appears to have decreased in a statistically significant way from 14.8% during ordinary activity to 12.5% during the COVID-19 emergency (*p* < 0.001).

Similarly, the proportion of traumatic patients assisted by the RD decreased from 16.6% during routine activity to 12.5% during the COVID-19 emergency (*p* < 0.001) (Table 6). Patients who could not be classified as traumatic or non-traumatic were excluded in this analysis. A statistically significant decrease in the number of diagnostic tests performed for trauma during the pandemic runtime was also found on a per week analysis (*p* < 0.001).

### 3.4. Poisson Regression Analysis

The effect of the lockdown period was modelled with the other variables of importance mentioned before, i.e., inpatients, outpatients, ED patients, and traumatic and not traumatic cases. As our study focused on the lockdown period, patient types, trauma, and their interactions were added to the null model so that the model of interest differed from the null one in the lockdown variable (year) and its interactions with the other variables. The model of interest is statistically much better than the null one (*p* < 0.001). The lockdown decreased the number of examinations significantly (*p* < 0.001). However, the effect was significantly lower for examinations performed on inpatients than for outpatients and ED patients (interaction between lockdown and patient type, *p* = 0.001). The interaction between lockdown and trauma was statistically highly significant, with examinations performed on trauma patients decreasing more than the others (*p* < 0.001). In summary, the Poisson regression confirmed the results of the simpler statistical tests.

## 4. Discussion

### 4.1. Immediate Impact of the COVID-19 Lockdown on the Radiology Department Workflow

Between 11 March 2020 and 26 May 2020, the overall diagnostic activity of the RD of our institution decreased significantly compared to the same period of the previous year, both considering the number of examinations and the number of assisted patients. A higher reduction in the imaging volume of the ERU is evident, while the GRU registered a more limited decrease. The number of performed diagnostic investigations per week and assisted patients per week confirmed a significant decrease during the reference period compared to the time-matched comparison.

Sorting data by patient type, the decrease in the number of diagnostic investigations performed affected all three categories: outpatients, inpatients, and ED patients. Compared to the same period of the previous year, the proportion of examinations performed on ED patients decreased significantly during the COVID-19 emergency, while the one relating to internal patients recorded a significant increase; finally, the proportion of examinations performed on outpatients did not change significantly between the two periods.

The increase in the proportion of investigations performed on inpatients may be explained in the first place by the fact that patients suffering from SARS-CoV-2 pneumonia who were hospitalized in COVID-19 wards may have required numerous investigations to monitor their pulmonary state during the course of the disease. Secondly, it must not be forgotten that inpatients generally require a high level of assistance and are subject to numerous diagnostic examinations during their stay at the hospital.

Furthermore, from 10 March 2020 to 18 May 2020, all routinary surgical activities were suspended in accordance with the directives of the local health authority. This measure was in line with those implemented by health systems in other regions impacted by the COVID-19 outbreak, where elective surgical activities were reduced to divert staff, beds, and other resources toward managing COVID-19 cases [22,23]. Surgeries performed in the reference period were not deferrable and needed considerable imaging support compared to what usually happens for minor surgeries.

The statistical analysis on the variation of the proportion of examinations performed for trauma allows us to state that following the COVID-19 outbreak, there was a significant decrease in these types of diagnostic investigations. Together with the reduction in the proportion of assisted ED patients, these data suggest that the fear of contagion may have driven part of the population to avoid seeking medical assistance for pathological conditions for which they would ordinarily go to the hospital. Additionally, lockdown measures may have limited the number of road and workplace accidents, as well as traumas related to sport or recreational activities, as other authors have stated [24,25]. This conclusion is also confirmed by analysing data relating to patients assisted for trauma; the proportion of the total number of patients was 16.5% during routine activity and decreased to 12.5% in the interval between 11 March and 26 May 2020.

The comparison of the number of trauma examinations per week between the COVID-19 period and the corresponding period of the previous year showed reductions of more than 50% for the first six weeks, while starting from week eight, the number of investigations shows a growth trend, approaching at the tenth and eleventh week the values of the time-matched comparison. This finding could be related to the fact that the climate of fear towards contagion gradually reduced during the lockdown, progressively pushing the population to act more in line with behaviours held in the previous year with a consequent change in the pattern of ED referrals [15]. The causes of the gradually greater sense of security can be traced back to the fact that the peak of new COVID-19 cases registered in Italy was reached and exceeded at the end of March 2020, with a consequent reduction after a few days in the number of deaths attributable to SARS-CoV-2 infections. Furthermore, it is reasonable to believe that the high media attention to the issues inherent to the pandemic and the commitment of the health authorities in raising awareness of the population contributed to changing the perception of the risks connected to COVID-19, alleviating the sense of “fear of the unknown” [26]. Finally, 18 May 2020 marked the beginning of the so-called “phase 2” of the management of the pandemic emergency in Italy and at that date, the local health authority of Ferrara resumed providing the health services that were suspended on 10 March 2020; even these events may have contributed to changing the attitude of citizens towards the use of hospital care.

Sorting data by method, all categories registered a decrease between the time-matched comparison and the reference period, except one; high-resolution CT scans (HRCT) were employed to a greater extent during the COVID-19 pandemic emergency. This increase is primarily attributable to the usefulness of the CT in aiding the management of SARS-CoV-2 pneumonia, both during the diagnostic phase and the monitoring of pulmonary parenchymal damage in affected patients [27]. Conversely, chest X-rays recorded a percentage decrease of their use during the time interval covered by the study, while their proportion to the total of the examinations decreased from 17.7% during routinary activity to 15% during the COVID-19 emergency. The explanation could be that chest X-rays do not appear to be indicated in the clinical management of COVID-19 as the evidence in the literature suggests that this method is not the best choice to identify lung changes caused by SARS-CoV-2 infection [28,29]. Also, chest X-rays are usually performed as a routine control tool in settings involving minor surgical treatments which were suspended during the pandemic emergency. These data do not agree with a Spanish multicenter study that found a preference for using conventional chest X-rays over chest CT as an initial diagnostic tool for COVID-19 [30].

Some of the most frequently used methods for investigating traumas have been selected and further surveyed.

While the number of brain MRIs and body MRIs performed for trauma did not vary significantly since they are not generally used in an acute setting, the data related to bone X-rays and ultrasounds are different.

Bone X-rays performed for trauma during the studied time interval almost halved compared to routine activity. The significant reduction could be because bone X-rays are commonly performed on patients who require assistance following minor traumas.

Body CTs were preferred over ultrasound scans on COVID-19 suspects to limit close contact between patients and operators.

On the other hand, brain CT scans are typically performed in contexts of severe traumas, such as road accidents. In the reference time frame, body CTs did not record significant changes: these findings may mean that in the province of Ferrara, most citizens received an adequate level of assistance following severe trauma, both because these types of patients usually require being brought to the hospital by ambulance or because, even despite the fear of SARS-CoV-2 contagion, some patients did not hesitate to seek help at the ED after a trauma which they considered potentially dangerous.

Our results are in line with the data collected by an Italian national survey which estimated a reduction in the radiological workflow of over 50% compared to the pre-pandemic period with a shift in examinations, particularly of non-COVID patients. This change resulted in radiologists’ fears of work overload for catching up on postponed tests with a delay in management in non-COVID patients [31].

A decrease in emergency radiological examinations was observed in several studies [32,33], particularly for trauma, reflecting the overall reduction in emergency room visits worldwide during the lockdown period [34,35,36]. As expected, political restrictions have reduced traffic accidents and accidents in other places, such as outdoor locations and workplaces [37,38,39]. 

In general, the climate of fear in the first phase of the pandemic and the consequent desire to maintain social distancing by avoiding contact with infected individuals could explain a significant decline in the use of hospital emergency services [40]. Indeed, several studies have shown that the fear of contagion, the adaptation of the healthcare system to the pandemic, and social restrictions imposed during the lockdown have led, for example, to a significant decrease in the rate of admissions for patients with acute cardiovascular disease, a reduction in the number of procedures, shortened periods of hospitalisation, and longer delays between the onset of the symptoms and hospital treatment [41]. This study has several limitations. First, analysing data in aggregate form does not allow us to evaluate in detail the variations of single types of investigations as some of them have been compressed into single large categories without considering the body segment investigated. Secondly, the clinical indications for which the examinations were performed have not been analysed except in the case of trauma. Additionally, the RIS-PACS used to manage the department’s activity and from which data were extracted, provides for a low degree of standardization regarding the compilation of the various fields in the software interface, making it difficult sometimes to sort patients into distinct referral categories. Finally, the data from this study only explores the experience of a hospital in one region of the country during an 11-week time frame of the COVID-19 pandemic. As discussed, the purpose of the study was to evaluate how the “immediate” impact of the stay-at-home orders included in the legislative decree of 3 September 2020 and the peak of infections in this period have changed the workflow of RD. The assessment of how the COVID-19 pandemic affected the activity in the following months is beyond the objectives of this study, also due to the difficulty of analysing subsequent waves that are heterogeneous in terms of vaccine availability, new investments in hospital resources, and the various legislative decrees governing them.

### 4.2. Lean Tools for Improving the Radiology Department Workflow

Considering these data, it is possible to hypothesize some key points based on the lean model to optimize resources, reduce waste, and improve results in RD workflow in future pandemics of COVID-19 or severe acute respiratory infections, particularly in periods of restriction of the citizens’ movements due to government decrees and fear of contagion:The significant decrease in the volume of emergency examinations during the global outbreak of coronavirus suggests a temporary downsizing of emergency radiological staff by relocating them to other tasks and to counter the financial strain deriving from COVID-19 patients on other radiology services, particularly those dedicated to the hospitalized patients.The increased volume of chest imaging in emergency settings may suggest using radiologists with expertise in thoracic imaging due to their increased sensitivity in detecting subtle pneumonia findings. In addition, specific training programs should be established for generalist radiologists who could also benefit from the support of artificial intelligence to improve the interpretation and efficiency of images, especially during night shifts [42].The excessive use of chest CT examinations and the concomitant decrease in conventional chest X-rays found in our data in the first wave of an unknown severe acute respiratory infection is not justified by guidelines regulating the use of imaging in COVID-19 in the subsequent pandemic phases. All major thoracic radiology societies advise against the indiscriminate use of imaging as a screening test for COVID-19 in patients with mild or no symptoms, while recommending its use based on symptom severity, pre-test probability, and COVID-19 testing [29,43,44].It is essential to separate the diagnostic pathways between suspected and non-suspected COVID-19 patients to prevent viral transmission between patients and healthcare workers [45]. One CT device should be closest to the COVID-19 emergency room only for infected patients and not too far away from the inpatient unit where patients with suspected or confirmed COVID-19 pneumonia are hospitalized.Mobile X-ray units and bedside ultrasounds should be encouraged to avoid the transportation of patients from the ward to the CT unit and to reduce the risk of contamination of staff and other patients. Although chest X-rays had low detection rates in the early stages of COVID-19 infection [27], these methods may be helpful in patient follow-up during treatment and also for detecting complications, such as pleural effusions or pneumothorax in mechanically ventilated subjects [46].For overcrowded RDs based on the local prevalence of COVID-19, non-urgent imaging exam appointments should be decreased or scheduled with a longer time gap, and accompanying visitors should be limited to avoid crowding the waiting zones.Since radiology is one of the medical specialties with a greater degree of digitalization, teleradiology and teleworking solutions should be strengthened in a similar dramatic scenario. The structured model of outsourced teleradiology has, in fact, been demonstrated to meet the requirements of emergency medicine during the pandemic with high diagnostic accuracy of chest CTs in the diagnosis of COVID-19 and a remarkable inter-observer agreement between teleradiologists with various degrees of experience and in contexts with different levels of disease prevalence [47]. By holding a small group of radiologists on-site and the rest of the group working safely from home to minimize the risk of cross-infection, teleradiology allows for the preservation of workload in the RD, increasing the productivity in other areas, such as administrative, operations, education, and research units, or updating strategies for optimizing workflow and safety protocols [48].

## 5. Conclusions

The COVID-19 outbreak had a tremendous impact on all radiological activities and particularly on investigations performed in the ED. This study can assist other health facilities to enrich the comprehension of the consequences of the COVID-19 pandemic on the activities of the RD and consequently, to make more effective choices in personnel management and operational protocol planning. Since the SARS-CoV-2 crisis is far from being completely resolved and the emergence of viral pathogens with strong epidemic potential is an event that tends to repeat itself, it is advisable that other institutions share their experiences, as comparative analyses will allow for obtaining a better understanding of the phenomenon, therefore guaranteeing better evidence-based decisions. In particular, current data based on the impact of social restriction measures on the workflow of healthcare institutions can guide policymakers in the event of a new COVID-19 peak or a new pandemic. These findings may suggest the reorganization of hospital activities, the possible reduction of trauma services, and the redistribution of staff to other health services. In these circumstances, it is helpful to redistribute the hospital’s financial resources to maintain adequate diagnostic care for all other diseases, such as cardiovascular or oncologic ones, avoiding diagnostic delays that could burden the health systems for years to come. Our study suggests the application of the Lean model to build new protocols for radiological practice in similar pandemic scenarios.

The results of its practical application need to be validated by further studies related to the improvement of radiological workflow and the containment of infection among healthcare workers and patients.

## Figures and Tables

**Figure 1 healthcare-10-01771-f001:**
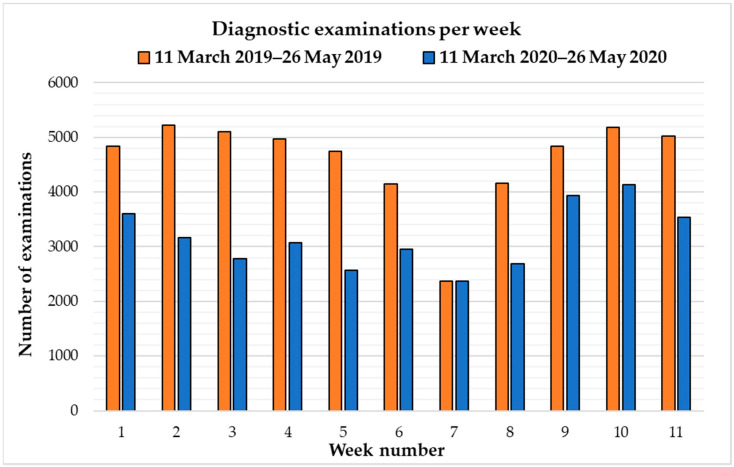
Graphical comparisons between the number of weekly diagnostic examinations performed during normal operativity (11 March 2019–26 May 2019, represented in orange) and the COVID-19 pandemic emergency period (11 March 2020–26 May 2020, represented in blue) in the radiology department of our hospital.

**Figure 2 healthcare-10-01771-f002:**
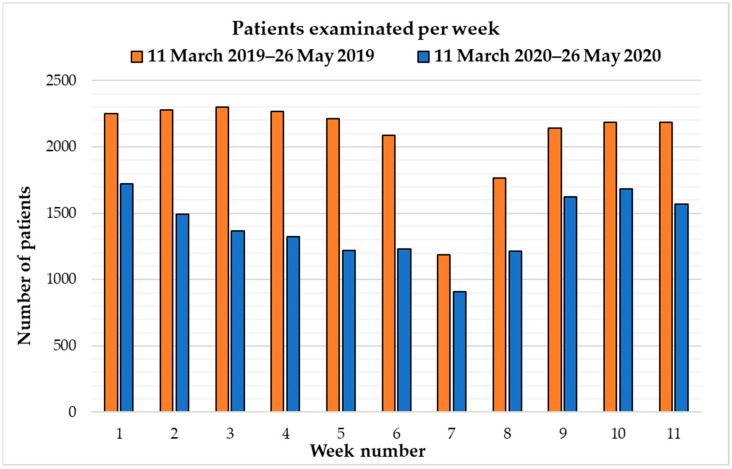
Graphical comparisons between the number of patients who underwent diagnostic examinations during normal operativity (11 March 2019–26 May 2019, represented in orange) and the COVID-19 pandemic emergency period (11 March 2020–26 May 2020, represented in blue) in the radiology department of our institution.

**Table 1 healthcare-10-01771-t001:** Number of patients assisted and diagnostic investigations performed during the ordinary activity period (11 March 2019–26 May 2019) and the pandemic emergency (11 March 2020–26 May 2020) in the radiology department of our institution; data are sorted relatively to the GRU and the ERU, and it is reported the proportion of examinations performed and patients assisted by each unit. The comparisons between the two-time intervals are expressed in absolute numbers and percentage changes.

	11 March 2019–26 May 2019		11 March 2020–26 May 2020		Comparison	
**Unit**	**Number of** **Patients**	**Proportion**	**Number of** **Patients**	**Proportion**	**Difference**	**Percentage**
GRU	10,866	55%	8070	60.6%	−2796	−25.7%
ERU	8905	45%	5252	39.4%	−3653	−41%
**Total**	19,771	-	13,322	-	−6449	−32.6%
**Unit**	**Number of** **Examinations**	**Proportion**	**Number of** **Examinations**	**Proportion**	**Difference**	**Percentage**
GRU	31,203	61.7%	23,260	66.8%	−7943	−25.5%
ERU	19,396	38.3%	11,552	33.2%	41	−40.4%
**Total**	50,599	-	34,812	-	−15,787	−31.2%

**Table 2 healthcare-10-01771-t002:** Number of diagnostic investigations performed per week within the ordinary activity (11 March 2019–26 May 2019) and during the COVID-19 pandemic emergency (11 March 2020–26 May 2020) by the radiology department of our institution; the comparisons between the two-time intervals are expressed both in absolute values and in percentages.

	11 March 2019–26 May 2019	11 March 2020–26 May 2020	Comparison	
Week Number	Number of Examinations	Number of Examinations	Difference	Percentage
Week 1 (11/03–17/03)	4841	3601	−1240	−25.6%
Week 2 (18/03–24/03)	5221	3171	−2050	−39.3%
Week 3 (25/03–31/03)	5109	2783	−2326	−45.5%
Week 4 (01/04–07/04)	4972	3070	−1902	−38.3%
Week 5 (08/04–14/04)	4750	2569	−2181	−45.9%
Week 6 (15/04–21/04)	4145	2957	−1188	−28.7%
Week 7 (22/04–28/04)	2364	2364	0	-
Week 8 (29/04–05/05)	4155	2687	−1468	−35.3%
Week 9 (06/05–12/05)	4840	3940	−900	−18.6%
Week 10 (13/05–19/05)	5181	4137	−1044	−20.2%
Week 11 (20/05–26/05)	5021	3533	−1488	−29.6%
**Total**	50,599	34,812	−15,787	−31.2%

**Table 3 healthcare-10-01771-t003:** Number of patients who underwent diagnostic examinations per week during the ordinary activity (11 March 2019–26 May 2019) and during the COVID-19 pandemic emergency (11 March 2020–26 May 2020) by the radiology department of our institution; the comparisons between the two-time intervals are expressed both in absolute values and in percentages.

	11 March 2019–26 May 2019	11 March 2020–26 May 2020	Comparison	
Week Number	Number of Patients	Number of Patients	Difference	Percentage
Week 1 (11/03–17/03)	2250	1721	−529	−23.5%
Week 2 (18/03–24/03)	2278	1492	−786	−34.5%
Week 3 (25/03–31/03)	2302	1366	−936	−40.7%
Week 4 (01/04–07/04)	2267	1321	−946	−41.7%
Week 5 (08/04–14/04)	2214	1221	−993	−44.9%
Week 6 (15/04–21/04)	2085	1230	−855	−41%
Week 7 (22/04–28/04)	1187	908	−279	−23.5%
Week 8 (29/04–05/05)	1765	1213	−552	−31.3%
Week 9 (06/05–12/05)	2140	1626	−514	−24%
Week 10 (13/05–19/05)	2187	1683	−504	−23%
Week 11 (20/05–26/05)	2187	1567	−620	−28.3%
**Total**	22,862	15,348	−7514	−32.9%

**Table 4 healthcare-10-01771-t004:** Number of patients and diagnostic investigations performed within the ordinary activity (11 March 2019–26 May 2019) and during the pandemic emergency (11 March 2020–26 May 2020) by the radiology department of our hospital; data are sorted relative to patient type, and the proportions for each category are reported. The comparisons between the two periods are expressed in absolute numbers and percentage changes.

	11 March 2019–26 May 2019		11 March 2020–26 May 2020		Comparison		
**Examinations** **by Patient Type**	**Number of** **Patients**	**Proportion**	**Number of** **Patients**	**Proportion**	**Difference**	**Percentage**	***p* Value**
Outpatient	7249	35.9%	4985	37.4%	−2264	−31.2%	0.004
Inpatient	4918	24.3%	3406	25.6%	−1512	−30.7%	0.012
ED patient	8035	39.8%	4929	37%	−3106	−38.7%	<0.001
**Total**	20,202	-	13,320	-	−6882	−34.1%	-
**Examinations** **by Patient Type**	**Number of** **Examinations**	**Proportion**	**Number of** **Examinations**	**Proportion**	**Difference**	**Percentage**	***p* Value**
Outpatient	15,179	30%	10,253	29.5%	−4926	−32.5%	>0.05
Inpatient	19,793	39.1%	14,393	41.3%	−5400	−27.3%	<0.001
ED patient	15,627	30.9%	10,166	29.2%	−5461	−34.9%	<0.001
**Total**	50,599	-	34,812	-	−15,787	−31.2%	-

**Table 5 healthcare-10-01771-t005:** Diagnostic examinations performed within the ordinary activity (11 March 2019–26 May 2019) and during the pandemic emergency (11 March 2020–26 May 2020) by the radiology department of our institution; data on examinations are sorted based on the clinical question for which their use was requested in trauma and not trauma. The comparisons between the two periods is expressed in absolute numbers and percentage changes.

	11 March 2019–26 May 2019		11 March 2020–26 May 2020		Comparison	
Examinations by Clinical Question	Number of Examinations	Proportion	Number of Examinations	Proportion	Difference	Percentage
Not trauma	43,008	85.2%	30,372	87.5%	−12,636	−29.4%
Trauma	7470	14.8%	4348	12.5%	−3122	−41.8%
Not determined	121	-	92	-	-	-
**Total**	50,478	-	34,720	-	−15,758	−31.2%

**Table 6 healthcare-10-01771-t006:** Number of trauma and not trauma patients who have undergone diagnostic investigations during the ordinary activity period (11 March 2019–26 May 2019) and the pandemic emergency (11 March 2020–26 May 2020) in the radiology department of our hospital. The comparisons between the two periods are expressed in absolute numbers and percentage changes.

	11 March 2019–26 May 2019		11 March 2020–26 May 2020		Comparison	
Patients by Clinical Condition	Number of Patients	Proportion	Number of Patients	Proportion	Difference	Percentage
Not trauma	15,316	83.4%	10,486	87.5%	−4830	−31.5%
Trauma	3038	16.6%	1495	12.5%	−1543	−50.8%
Not determined	52	-	49	-		-
**Total**	18,354	-	11,981	-	−6373	−34.7%

## Data Availability

The data presented in this study are available on request from the corresponding author. The data are not publicly available due to privacy restrictions.

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
