# Peer review of "Best Practices on Radiology Department Workflow: Tips from the Impact of the COVID-19 Lockdown on an Italian University Hospital"

_healthcare, 2022, doi:10.3390/healthcare10091771_

Round 1
Reviewer 1 Report (New Reviewer)
Thank you for conducting this poetically important study.
The revised thesis is logical and well-organized.
This study can assist other health facilities to enrich the comprehension of the consequences of the COVID-19 pandemic on the activities of the RD and, consequently, to make more effective choices in personnel management and operational protocols planning.
Author Response
We thank the reviewer for his precious support in reviewing the article.
Reviewer 2 Report (New Reviewer)
In this study, the authors have analyzed the impact of the COVID-19 pandemic on the imaging examination volume in a radiology department in Northern Italy. They found that the pandemic has significantly reduced total imaging examination volume.
In general the paper is well written and provides an interesting overview of the impact of COVID-19 pandemic on a radiology department.
I have the following comments that the authors should address.
1. In general, the authors should use the active or passive voice consistently but not one of these alternatively
2. The authors should not use colloquial terms such as “exams”, “carried out” but use “examination” and “performed” in the whole manuscript.
3. In general, there is no need for reporting P values with 5 digits. P values should be rounded at 3 digits.
Abstract
4. Please give actual P values instead of P < 0.05
Introduction
5. The authors should add the following reference to document the impact of COVID-19 on interventional radiology activities
Denys A, Guiu B, Chevallier P, Digklia A, de Kerviler E, de Baere T. Interventional oncology at the time of COVID-19 pandemic: Problems and solutions. Diagn Interv Imaging 2020;101(6):347-353. doi: 10.1016/j.diii.2020.04.005.
6. The last section should be a sentence starting with “The purpose of this study was to ….”
7. The 1.1 section should be better placed in M&M section
Discussion
8. The two sections of Discussion should be 5.1 and 5.2 (5.1 should be renumbered 5.2)
The section 5.1. should have more references. The following references should be added
Li J, Long X, Wang X, Fang F, Lv X, Zhang D, Sun Y, Hu S, Lin Z, Xiong N. Radiology indispensable for tracking COVID-19. Diagn Interv Imaging 2021;102(2):69-75. doi: 10.1016/j.diii.2020.11.008.
lquézar-Arbé A, Piñera P, Jacob J, Martín A, Jiménez S, Llorens P, Martín-Sánchez FJ, Burillo-Putze G, García-Lamberechts EJ, González Del Castillo J, Rizzi M, Agudo Villa T, Haro A, Martín Díaz N, Miró Ò. Impact of the COVID-19 pandemic on hospital emergency departments: results of a survey of departments in 2020 - the Spanish ENCOVUR study. Emergencias 2020;32(5):320-331.
Gil-Rodrigo A, Miró Ò, Piñera P, Burillo-Putze G, Jiménez S, Martín A, Martín-Sánchez FJ, Jacob J, Guardiola JM, García-Lamberechts EJ, Espinosa B, Martín Mojarro E, González Tejera M, Serrano L, Agüera C, Soy E, Llauger L, Juan MÁ, Palau A, Del Arco C, Rodríguez Miranda B, Maza Vera MT, Martín Quirós A, Tejada de Los Santos L, Ruiz de Lobera N, Iglesias Vela M, Torres Garate R, Alquézar-Arbé A, González Del Castillo J, Llorens P; en representación de la red de investigación SIESTA. Analysis of clinical characteristics and outcomes in patients with COVID-19 based on a series of 1000 patients treated in Spanish emergency departments. Emergencias 2020;32(4):233-241
Nuñez JH, Sallent A, Lakhani K, Guerra-Farfan E, Vidal N, Ekhtiari S, Minguell J. Impact of the COVID-19 Pandemic on an Emergency Traumatology Service: Experience at a Tertiary Trauma Centre in Spain. Injury 2020;51(7):1414-1418. doi: 10.1016/j.injury.2020.05.016.
The authors should briefly compare what they did to what has been done in other countries
9. In this paragraph “All major thoracic radiology societies advise against the indiscriminate use of imaging as a screening test for COVID-19 in patients with mild or no symptoms while recommending its use based on symptom severity, pre-test probabil ity and COVID-19 testing” the authors should add the following reference from the Italian Society
Neri E, Miele V, Coppola F, Grassi R. Use of CT and artificial intelligence in suspected or COVID-19 positive patients: statement of the Italian Society of Medical and Interventional Radiology. Radiol Med 2020;125(5):505-508. doi: 10.1007/s11547-020-01197-9
10. In this paragraph “it is essential to separate the diagnostic pathways between suspected and non-suspected COVID-19 patients to prevent viral transmission between patients and healthcare workers. One CT device should be closest to the COVID-19 emergency room only for infected patients and not too far away from the inpatient unit where patients with suspected or confirmed COVID-19 pneumonia are hospitalized.” This authors should add the following reference that documents the separation of the different pathways.
Kato S, Ishiwata Y, Aoki R, Iwasawa T, Hagiwara E, Ogura T, Utsunomiya D. Imaging of COVID-19: An update of current evidences. Diagn Interv Imaging 2021;102(9):493-500. doi: 10.1016/j.diii.2021.05.006.
11. In this paragraph “mobile X-ray units and bedside ultrasounds should be encouraged to avoid the transportation of patients from the ward to the CT unit and to reduce the risk of contamination of staff and other patients. These methods may be helpful in patient follow-up during treatment and also for detecting complications such as pleural effusions or pneumothorax in mechanically ventilated subjects” the authors should acknowledge the modest sensitivity and specificity of chest X-ray compared to CT and add the following reference
Li J, Long X, Wang X, Fang F, Lv X, Zhang D, Sun Y, Hu S, Lin Z, Xiong N. Radiology indispensable for tracking COVID-19. Diagn Interv Imaging 2021;102(2):69-75. doi: 10.1016/j.diii.2020.11.008.
12. In this paragraph, “since Radiology is one of the medical specialities with a greater degree of digitalization, teleradiology and teleworking solutions should be strengthened in a similar dramatic scenario. Teleradiology by holding a small group of radiologists on-site and the rest of the group working safely from home to minimize the risk of cross-infection allows for 439 the preservation of workload in the RD, increasing productivity in other areas such as administrative, operations, education, and research units or updating strategies for optimizing workflow and safety protocols”
The authors should add this relevant recent reference
Nivet H, Crombé A, Schuster P, Ayoub T, Pourriol L, Favard N, Chazot A, Alonzo-Lacroix F, Youssof E, Ben Cheikh A, Balique J, Porta B, Petitpierre F, Bouquet G, Mastier C, Bratan F, Bergerot JF, Thomson V, Banaste N, Gorincour G. The accuracy of teleradiologists in diagnosing COVID-19 based on a French multicentric emergency cohort. Eur Radiol 2021;31(5):2833-2844. doi: 10.1007/s00330-020-07345-z.
13. References
Refs 39 is the same than ref 25, except that it is the Chest version instead or Radiology version of the paper. I suggest deleting ref. 39
Author Response
We thank the Reviewer for the careful attention dedicated to our manuscript and for all the insightful indications. We believe that our revised manuscript has importantly improved by following the Reviewer’s indications.
- We thank the Reviewer for this valuable suggestion. Respecting the indications of the Reviewer, we have modified the text in many points to standardize it in passive form. Changed sentences are marked up using the "Track Changes" function.
- We thank the Reviewer for this valuable suggestion. To comply with the Reviewer's indications, we have changed the suggested words in the text, figures and tables (also in supplementary materials). Changed terms are marked up using the "Track Changes" function.
- We apologize for the inaccuracy and have rounded P values at 3 digits(also in supplementary materials).
- We thank you for the valuable indication: we have used the actual P values in the abstract.
- We have added the suggested reference [14] in the "Introduction".
- The last section of the "Introduction" now starts with the sentence "The purpose of this study was to…."(line 82).
- Thanks for this helpful tip. Section 1.1 has been placed in section M&M (now numbered 2.1). In M&M, the title "2.2. Source and population of data" has been introduced, and the paragraph "2.3 Statistical analysis" has been moved to this section (old version in section 3).
- With the numbering changes previously explained, the discussion section has been split into sections 4.1 (now titled 4.1. Immediate impact of the COVID-19 lockdown on the radiology department workflow) and 4.2. In the current sub-section 4.1 the suggested bibliographic references have been added [22-23-27-30]. A brief comparison with what happened in other countries was made (lines 376-378).
- The suggested reference[44] was added in the paragraph Discussion(line 456).
- The suggested reference[45] was added in the paragraph Discussion(line 459).
- The suggested reference[27] was added in the paragraph Discussion(line 368).
- The suggested reference[47] was added in the paragraph Discussion (line 477).
- We deleted reference 39 (Chest version). The Radiology version of the paper is now numbered 29.
Reviewer 3 Report (Previous Reviewer 1)
Thank you. No further comments.
Author Response
We thank the reviewer for his attention and expertise in reviewing the article.
This manuscript is a resubmission of an earlier submission. The following is a list of the peer review reports and author responses from that submission.
Round 1
Reviewer 1 Report
I have the following comments:
-In the Abstract (line 21), please define the acronym 'ERU', which is first defined later on in the article body.
Moreover (lines 24-26), please provide quantitative data for the decreased number of emergency patients assisted by the RD during the COVID-19 outbreak, along with the p-value of its related comparison with pre-COVID-19 data.
-Introduction (line 46). Please replace the Italian term 'Sistema Sanitario Nazionale' with 'Italian public healthcare system' for better understandability by international readers.
Furthermore, please name the 'Arcispedale Sant'Anna in Ferrara' in full only once in the text, and then replace the full name with a simpler wording such as 'our Institution' or 'our hospital'.
-In the Discussion section, it would be important to compare the study findings with those from the Italian nationwide survey by Coppola F et al [Insights Imaging. 2021 Feb 17;12(1):23. doi: 10.1186/s13244-021-00962-2], which gives insights on the impact of the COVID-19 pandemic on the professional activity of Italian radiologists, and more generally on their professional and personal wellbeing.
I would also suggest stressing which specific advancements to knowledge are brought by the study findings, which are in line with several other articles published in the literature.
-Introduction (line 271). Please move the year '2020' at the end of the sentence.
-Tables. Why are numbers highlighted in red?
Author Response
Response to Reviewer 1 Comments
We thank the Reviewer for the careful attention dedicated to our manuscript and for all the insightful indications. We believe that our revised manuscript has importantly improved by following the Reviewer’s indications.
Point 1: In the Abstract (line 21), please define the acronym 'ERU', which is first defined later on in the article body.
Moreover (lines 24-26), please provide quantitative data for the decreased number of emergency patients assisted by the RD during the COVID-19 outbreak, along with the p-value of its related comparison with pre-COVID-19 data.
Response 1: We thank the Reviewer for these precious suggestions. In order to comply with the Reviewer’s indications, we defined the acronym ‘ERU’ in the Abstract (line 21 of revisited manuscript) and we added quantitative data for the decreased number of emergency patients assisted by the RD during the COVID-19 outbreak, along with the p-value of its related comparison with pre-COVID-19 data (lines 25,26 of revisited manuscript).
Point 2: Introduction (line 46). Please replace the Italian term 'Sistema Sanitario Nazionale' with ' Italian public healthcare system ' for better understandability by international readers. Furthermore, please name the 'Arcispedale Sant'Anna in Ferrara' in full only once in the text, and then replace the full name with a simpler wording such as 'our Institution' or 'our hospital'.
Response 2: We thank the Reviewer for these helpful indications. According to the Reviewer’s indications, we replaced the Italian term 'Sistema Sanitario Nazionale' with 'Italian public healthcare system' in the Introduction section (line 46). Moreover, according to the indications of the Reviewer, we have appointed 'Arcispedale Sant'Anna in Ferrara' in full only once in the text (University Hospital “Arcispedale Sant’Anna” paragraph), and then we replaced the full name with the suggested terms 'our Institution' or 'our hospital' in the subsequent sections of the article and in the figures and tables captions. Changed terms are marked using the “Trace Changes" function.
Point 3: In the Discussion section, it would be important to compare the study findings with those from the Italian nationwide survey by Coppola F et al [Insights Imaging. 2021 Feb 17;12(1):23. doi: 10.1186/s13244-021-00962-2], which gives insights on the impact of the COVID-19 pandemic on the professional activity of Italian radiologists, and more generally on their professional and personal wellbeing.
I would also suggest stressing which specific advancements to knowledge are brought by the study findings, which are in line with several other articles published in the literature.
Response 3: we thank the Reviewer for the opportunity to compare our data with the results of an Italian national survey by adding an important reference. To comply with the Auditor's indication, we added this missing reference in the final part of the discussion (lines 348-352) by comparing our data with those of the Italian national survey.
Furthermore, to underline the specific advances in knowledge brought by our data, we have expanded the conclusions section also following the indications of the Academic Editor (lines 392-400).
Point 4: Introduction (line 271). Please move the year '2020' at the end of the sentence.
Response 4:We thank the Reviewer for careful rectification. According to the Referee's instructions, we have moved the indication of the year "2020" to the end of the sentence (line 296 of revisited manuscript).
Point 5: Tables. Why are numbers highlighted in red?
Response 5: Thanks to the Reviewer for pointing out this formatting error. In line with the Reviewer's comment, we have restored the color of the data in the tables to black.

Reviewer 2 Report
Manuscript ID: Healthcare - 1788724
Title: The immediate impact of the COVID-19 lockdown on emergency radiology workflow: evidence from an Italian University Hospital
Referee’s comments
A. General comment
The present paper attempts to demonstrate the impact of COVID-19 lockdown on emergency radiology workflow for a specific Italian University Hospital. The article presents clinical data and could be interesting in order to get an idea of COVID-19 effect on medical imaging examinations, however I propose the rejection of the present manuscript for the following reasons:
1) To my opinion, the duration period (11/03/2020 - 26/05/2020) is too short to make a conclusion of the data analysis. Maybe a two-year duration 03/2022 – 05/2022 could capture better the impact of COVID-19 on emergency radiology workflow. In addition, that period there was a global change in every daily action, so it is normal to have an impact on medical imaging examinations as well. A two-year duration would help to have a clearer image of COVID-19 effect on the radiology workflow of the Hospital.
2) Data are provided for a specific hospital. The manuscript refers to a specific hospital (a case study) which is limited to the general impact of COVID-19 on emergency radiology. A total number of data from several hospitals would strengthen the soundness of the study.
3) To my opinion, the manuscript seems to suit better as a technical report rather than a paper.
Author Response
Response to Reviewer 2 Comments
General comment
The present paper attempts to demonstrate the impact of COVID-19 lockdown on emergency radiology workflow for a specific Italian University Hospital. The article presents clinical data and could be interesting in order to get an idea of COVID-19 effect on medical imaging examinations, however I propose the rejection of the present manuscript for the following reasons:
Point 1: To my opinion, the duration period (11/03/2020 - 26/05/2020) is too short to make a conclusion of the data analysis. Maybe a two-year duration 03/2022 – 05/2022 could capture better the impact of COVID-19 on emergency radiology workflow. In addition, that period there was a global change in every daily action, so it is normal to have an impact on medical imaging examinations as well. A two-year duration would help to have a clearer image of COVID-19 effect on the radiology workflow of the Hospital.
Response 1: We thank the Reviewer for his interesting comments. First, the data from this study refer to the experience of a hospital in one region of the country in a period of 11 weeks since the Covid-19 pandemic. The purpose of the work, as the title suggests, was to evaluate the "immediate" impact of the lockdown on the activity of the radiology department. We have analyzed the time frame in which the massive restrictive measures inserted in the DCPM 9/03/2020, which marked the beginning of the Italian "lockdown" together with the citizens' fear of the peak of infections in this period, have changed the access to the radiology department. The evaluation of how the COVID-19 pandemic affected the activity in the following months is beyond the objectives of this work, also due to the difficulty of analysing subsequent waves, heterogeneous in terms of vaccine availability and new investments in hospital resources and because various legislative decrees governed them. The suggested extension of the study period to a two-year duration could have better highlighted the impact of COVID-19 on radiology workflow, but the acquisition of new machinery in the year 2021 (a new CT and MRI in addition to the previous ones) have certainly changed the radiology workflows, making the year 2022 not comparable with the two periods analyzed in our study in which instead the same number of scanners were used. Thanks for your suggestion, it could be the starting point for further work. We have included this reflection within the limits of the work, on the recommendation of the Academic Editor (lines 372-380 of revisited manuscript).
Point 2: Data are provided for a specific hospital. The manuscript refers to a specific hospital (a case study) which is limited to the general impact of COVID-19 on emergency radiology. A total number of data from several hospitals would strengthen the soundness of the study.
Response 2: Thanks for the suggestion. This is a monocentric study: the Sant'Anna Arcispedale is in fact a tertial academic referral center of the province of Ferrara. It would also be interesting to evaluate, also through meta-analysis, the existing evidence in the literature and coming from different centers. We hope that the data we provided will be available for other studies.
Point 3: To my opinion, the manuscript seems to suit better as a technical report rather than a paper.
Response 3: We thank the reviewer for the indication. Our manuscript does not want to be a technical report but a descriptive analysis of the activity of a radiology department of a single hospital in a specific pandemic phase. By quantifying the change in imaging use experienced in our radiology department, we hope that similar organizations will be able to extrapolate the effects of the crisis on their institutions. Institutions with diverse backgrounds should be encouraged to disclose their data to show a more accurate picture of how this crisis has changed clinical practices and how we can better prepare for the next pandemic.

Reviewer 3 Report
Fabio Pellegrino and co-workers report in this article “The immediate impact of the COVID-19 lockdown on emergency radiology workflow: evidence from an Italian University Hospital”. In this article. The authors summarized the effect of the COVID 19 outbreak's impact on all the radiological activities at the university hospital in northern Italy. The data showed that there was a decrease in all the diagnoses in the radiology department during the COVID 19 outbreak compared to the previous year.
In summary, the study was organized and presented well. Therefore, I recommend the editor accept the manuscript.
Author Response
Response to Reviewer 3 Comments
Fabio Pellegrino and co-workers report in this article “The immediate impact of the COVID-19 lockdown on emergency radiology workflow: evidence from an Italian University Hospital”. In this article. The authors summarized the effect of the COVID 19 outbreak's impact on all the radiological activities at the university hospital in northern Italy. The data showed that there was a decrease in all the diagnoses in the radiology department during the COVID 19 outbreak compared to the previous year.
In summary, the study was organized and presented well. Therefore, I recommend the editor accept the manuscript.
We are grateful to the Reviewer for having assessed our article with his experience in the field and for having judged it positively.

Round 2
Reviewer 2 Report
I suggest the rejection of the manuscript for the reasons explained in my previous initial report.